

# Bathymetry and oceanic flow structure at two deep passages crossing the Lomonosov Ridge

Göran Björk[1], Martin Jakobsson[2], Karen Assmann[1], Leif Andersson[1], Johan Nilsson[3,4], Christian Stranne[2,5], and Larry Mayer[5]

[1]Department of Marine Sciences, University of Gothenburg, Gothenburg 405 30, Sweden

[2]Department of Geological Sciences, Stockholm University, Stockholm 106 91, Sweden.

[3]Bolin Centre for Climate Research, Stockholm University, Stockholm106 91, Sweden.

[4]Department of Meteorology, Stockholm University, Stockholm 106 91, Sweden.

[5]Center for Coastal and Ocean Mapping, University of New Hampshire, New Hampshire 03824, USA

*Correspondence to:* Göran Björk (goran.bjork@marine.gu.se)

**Abstract.** The Lomonosov Ridge represents a major topographical feature in the Arctic Ocean which
has a large effect on the water circulation and the distribution of water properties. This study presents
detailed bathymetric survey data along with hydrographic data at two deep passages across the ridge:
A southern passage (80-81 °N) where the ridge crest meets the Siberian continental slope and a
northern passage around 84.5 °N. The southern channel is characterized by smooth and flat bathymetry
around 1600-1700 m with a sill depth slightly shallower than 1700 m. A hydrographic section across
the channel reveals an eastward flow with Amundsen Basin properties in the southern part and a
westward flow of Makarov Basin properties in the northern part. The northern passage includes an
approximately 72 km long and 33 km wide trough which forms an intra basin in the Lomonosov Ridge
morphology (the Oden Trough). The eastern side of Oden Trough is enclosed by a narrow and steep
ridge rising 500-600 m above a generally 1600 m deep trough bottom. The deepest passage (the sill) is
1470 m deep and located on this ridge. Hydrographic data show irregular temperature and salinity
profiles indicating that water exchange occurs as midwater intrusions bringing water properties from
each side of the ridge in well-defined but irregular layers. There is also morphological evidence that
some rather energetic flows may occur in the vicinity of the sill. A well expressed deepening near the
sill may be the result of seabed erosion by bottom currents.

**1 Introduction**

The Arctic Ocean includes the northernmost loop of the global ocean circulation system. Warm water
from the North Atlantic flows through Fram Strait between Svalbard and Greenland and continues into
the central Arctic Ocean basin as a warm core along the northern Barents Sea shelf slope (Fig. 1).
Another branch of Atlantic water flows across the Barents Sea where it undergoes significant cooling
by heat loss to the atmosphere before entering the central Arctic Ocean further east in the St. Anna
Trough (Schauer et al., 2002). The two branches of Atlantic inflow meet to partially mix together and




continue further eastward along the continental shelf slopes of Kara, Laptev and East Siberia seas
(Rudels et al., 2000). The inflow of Atlantic water implies a net heat transport from low to high
latitudes and results in the entire central Arctic Ocean basin filling up with a relatively warm (>0 $^0$C)
layer at depths between about 100 and 600 m, the Atlantic layer. The maximum temperature core is
around 200-300 m. The temperature of the Atlantic water inflow has not been steady, not even over
short time-scales. Several warm pulses on decadal scales have been observed as well as sudden
changes in the core temperature (Dmitrenko et al., 2009; Woodgate et al., 2001).

The currents in the central Arctic Ocean are generalized by a weak interior circulation and intensified
boundary currents following the slopes of the shallow continental shelf seas that nearly enclose the
central basin. The boundary currents carry not only the warm core of the Atlantic layer (>0 $^0$C) but
also intermediate and deep water with lower temperature that generally decreases with depth. The
system of boundary currents including water from the two branches of Atlantic inflow is referred to as
the Arctic Circumpolar Boundary Current (Rudels et al., 1999). However, the detailed circulation is
far from one simple flow along the continental slopes of the shelves which emerges from spatial
contrasts of the Atlantic layer and deep water properties (Rudels et al., 2012). The complex seafloor
landscape consists of extensive submarine ridges with morphologies that influence and steer the
currents (Fig. 1).

As is the general case for the mean circulation in the weakly stratified high-latitude oceans, the
circulation in the Arctic is strongly guided by bathymetry with currents tending to follow isobaths and
giving rise to several internal circulation loops associated with deep sea ridges (see e.g., Nøst and
Isachsen, 2003). At places where the deep sea ridges meet the continental slope they may deflect a part
of the boundary current from the continental slope and make it follow the ridge instead. The largest
deep sea ridge system in the Arctic Ocean is the Lomonosov Ridge (LR) rising up several thousand
meters above the abyssal plains and stretching all the way from the Siberian slope to the continental
slope north of Greenland (Fig. 1). The LR also defines the border between the two major ocean basins
in the Arctic: the Eurasian Basin, with sub-basins Nansen Basin and Amundsen Basin and the
Amerasian Basin, with sub-basins Makarov Basin and Canada Basin. The impact of the LR as a major
obstacle for the boundary current's passage along the Siberian slope from the Eurasian Basin to the
Amerasian Basin side is clearly seen in hydrographic data (Anderson et al., 1994), current
measurements (Woodgate et al., 2001) and ocean modelling studies (Aksenov et al., 2001). The
boundary current splits up in two branches where the LR meets the Siberian shelf slope (Fig. 1). One
branch turns north and flows along the ridge towards Greenland and one part passes the ridge and
continues along the shelf slope in the Amerasian Basin. In addition to the passage of warm Atlantic
water flowing across the LR from the Amundsen Basin to the Makarov Basin, there is also evidence
based on mooring data that deep water from the Makarov Basin passes the ridge in the opposite
direction in this area (Woodgate, 2001).



Mapping the detail pathways of the warmer water of Atlantic origin is of importance as the heat it brings influences the Arctic Ocean environment, specifically the marine cryosphere, including for example gas hydrates stored in bottom sediments, sea ice and outlet glaciers. More specifically, the inflowing warm Atlantic water controls the temperature near the seabed along the continental shelf

slope, which affects the location of the Gas Hydrate Stability Zone and thus impacts the storage and release of methane (Biastoch et al., 2011; Stranne et al., 2016; Westbrook et al., 2009). Furthermore, the heat carried with the inflow has a large potential to melt the perennial sea ice cover (Polyakov et al., 2010), although mostly in areas with weak salinity stratification such as north of Svalbard. In most parts of the Arctic, the Atlantic layer is shielded from the sea ice by a strong cold halocline resulting

from freshwater supply by rivers and low salinity water coming from the north Pacific through Bering Strait (Sirevaag and Fer, 2009; Linders and Björk, 2013). Another example illustrating the complex pathways of Atlantic water is that it finds its way across Lincoln Sea to reach the Petermann Fjord of northwestern Greenland where the heat it brings causes melting of the underside of Petermann Glacier's floating ice tongue (Münchow et al., 2007).

The bathymetric portrayal of the LR in the latest version 3.0 of the International Bathymetric Chart of the Arctic Ocean (IBCAO) is mainly based on sparse single beam echo soundings from icebreakers and submarines and digitized depth contours from published maps, apart from a few areas mapped with multibeam echo sounder (Jakobsson et al., 2012). The sparse source data implies that bathymetric details of importance from an oceanographic perspective may be missed in some areas, such as the

location of bathymetric passages or saddles in LR, which are critical as this is where a large part of the water exchange between the basins can occur. The (presumably) deepest passage across the LR, with a sill depth of 1870 m, has been located at $88^0$ N. This passage was mapped in detail with multibeam together with hydrographic observations in 2005 and it was shown that a substantial flow of Canadian Basin Deep Water passes through it (Björk et al., 2007; Björk et al., 2010).

Here we present a detailed bathymetry of two other deep passages in the LR morphology together with hydrographic data acquired during the SWERUS-C3 (Swedish-Russian-US Arctic Ocean Investigation of Climate-Cryosphere-Carbon Interactions) expedition in 2014 with Swedish icebreaker (IB) *Oden*. The southernmost of these passages is located close to the Siberian shelf slope between about 80°N and 81°N where it can be expected that most of the Atlantic water passes the LR towards the

Amerasian Basin. The other passage is located further north at ~85°N in an area that has not been studied before but has been suggested as a possible location for exchange of Atlantic water (Woodgate et al. , 2001). Our study shows the critical importance of knowing the detailed shape of the seafloor in order to map the bottom currents, which in the Arctic Ocean are critical for the distribution of heat and other properties and have implications for the marine environment, including the vulnerable marine

cryosphere.



## 2 Methods

CTD observations were made using a SeaBird 911 CTD equipped with dual SeaBird temperature
(SBE 3), conductivity (SBE 04C) and oxygen sensors (SBE 43). Salinity samples were analyzed using
a Guildline Autosal instrument which was kept in a well isolated lab container with relatively constant

temperature. The salinometer was calibrated using one standard sea water ampule (IAPSO standard
sea water from OSIL Environmental Instruments and Systems) before each batch of 24 samples. The
CTD data files were post processed with standard SeaBird data processing software. They were
despiked manually by first visually identifying spikes and then interpolating across the spike in the file
with full time resolution. The alignment parameter was tuned for each sensor pack following the

suggested method described in the SeaBird Data Processing manual. Salinities were compared with
bottle salinities analyzed with the Autosal lab salinometer. This analysis revealed a systematic offset
of conductivity sensor 1, from station 16 and onward, corresponding to about 0.002 PSU higher
salinities compared to the salinometer and to conductivity sensor 2. A conductivity slope correction
was calculated and the raw data conversion was redone with the slope correction.

A description of the geophysical mapping program onboard IB *Oden* during the SWERUS-C3 2014
expedition is included in Jakobsson et al. (2016). Here we provide a methods summary with emphasis
on the mapping of the areas of the LR that are the focus for this study. Multibeam swath bathymetry
was acquired using the Kongsberg EM 122 (12 kHz, 1°x1°) multibeam system installed in IB *Oden*.

The CTD observations gathered for the oceanographic program were used for sound velocity control
and supplemented with XBT (Expendable Bathy Thermograph) casts. The target areas of the LR were
bathymetrically mapped with between 50 and 100 % overlapping multibeam swaths to acquire a high
quality imagery of the ridge morphology. All multibeam bathymetric data were processed using a
combination of the software Caris and Fledermaus-QPS. Grids with horizontal resolutions ranging

between 15x15 m and 30x30 m were produced and used for the final analyzes in the 3D environment
of Fledermaus and in the GIS software ArcMap. In addition to multibeam bathymetry, sub-bottom
profiles were collected using the Kongsberg SBP 120 3°x3° chirp sonar installed in IB *Oden*. The
chirp sonar was operated continuously using a 2.5-7 kHz pulse.

The high quality of the bathymetric data collected was due, in large part, to the exceptionally ice free

conditions along the LR during summer 2014.

## 3 Results

### 3.1 Southern passage



The LR meets the continental slope of the Siberian shelf approximately at the defined border between the Laptev and East Siberian Seas is (Fig. 1). This section of the LR, here referred to as the southern passage, is characterized by smooth and flat bathymetry with depths around 1600–1700 m in the deeper central parts where the hydrographic section is located (Fig. 2 a, c). North of the deepest flat part, the ridge rises rather abruptly to reach depths shallower than 800 m. The southern end of the southern passage is confined by the continental slope that rises to the continental shelf. The single multibeam track on the western side of the surveyed section reveals a stripe of rugged bottom topography with generally shallower depths than in the central parts. Increasing depths at the southern end of this western section suggest that the bathymetric sill of the southern gap is likely located at the south-western rim of the central plateau. However, since the southern passage in the LR not was completely covered by the SWERUS-C3 multibeam survey, it is not possible to exactly pinpoint the location of the sill. We cannot exclude that there might be a shallower area forming a sill where there is no multibeam coverage between the hydrographic section and the western single track of multibeam data. However, the multibeam data we have at hand together with the general morphology of the LR provided by Version 3.0 of IBCAO, suggest a sill shallower than 1700 m somewhere south of 81 15'N 141 30'E (Fig. 2a). IBCAO is generally in good agreement with the newly acquired multibeam bathymetry, but approximately 50 m shallower in the deeper parts of the southern passage than our new multibeam bathymetry. IBCAO Version 3.0 is in this area primarily based on digitized contours from the Russian bottom relief map of the Arctic Ocean (Naryshkin, 2001), which do not include information on the bathymetric source data used.

The section across the passage (along the ridge crest) close to the Siberian continental slope (Fig. 3) shows a low salinity surface layer down to about 20 m with high temperature (up to 3 °C) above a cold layer close to the freezing point. The low salinity is likely a result of ice melting and possibly also fresh water input from the north Siberian rivers. The high surface temperature is a result of the open water conditions which allowed for heat supply by absorption of solar radiation and air-sea heat exchange during the summer. The interval between about 100 m and 900 m contains the Atlantic water layer with temperature above 0 °C. The highest temperature in the Atlantic water core is above 1.5 °C and is seen in the southern half of the section indicating that Atlantic water from the Amundsen Basin passes the ridge through this channel with a high temperature/salinity core in the southern part. The deep waters (>1000 m) are characterized by decreasing temperature and increasing salinity towards the bottom. The deepest part of the section shows anomalous properties in a thin bottom layer with higher salinity and temperature. The density (not shown) is also higher in this layer since the salinity dominates the effect on the density. Another noteworthy feature is station 137 with higher temperature and salinity at mid depth in the range 600-1300 m. This appears to be to be a small anticyclonic eddy which is not fully resolved with the present station spacing which is larger than the Rossby radius of ~7.5 km based on the first baroclinic mode phase speed of station 137. The oxygen





section shows generally high concentration in the surface layer roughly matching the salinity distribution. It is relatively constant in most of the depth interval 200-1000 m with concentration around 295 μmol kg$^{-1}$. The deepest part shows lower concentrations at the northern stations below 1200 m. The concentration is also significantly lower in the thin bottom layer matching the salinity

and temperature data.

The TS profiles are compared with reference stations in the Makarov Basin (Stn. 145) and Amundsen Basin (Stn. 148) in figure 4. It can be seen that the three northernmost stations are dominated by Makarov Basin TS properties (red curves) with less saline and colder water in the depth interval 700-1300 m. The other stations in the section follow more the Amundsen Basin reference profile. The

situation is more complicated in the Atlantic Layer, because of the eddy-like feature at stations 137 and 138 around 500 m depth with significantly lower salinity and slightly colder temperature. Also the southernmost station (Stn. 143) has a similar pattern in the Atlantic layer. Apart from these three stations it is clear that the Makarov Basin properties dominate the three northern stations above a depth of about 1500 m and well into the Atlantic layer. This is also seen in the oxygen data with lower

concentrations in the northern deep part of the section indicating a flow from the Makarov Basin having generally lower oxygen concentration in this depth interval (see also Fig. 5). These observations thus suggest a flow from the Amundsen Basin towards the Makarov Basin in the depth interval 400-1300 m over the major southern part of the channel, but there is also clear evidence of a flow in the opposite direction in the northern part.

Deeper down in water column, below 1500 m, the salinity and temperature increases sharply in a bottom layer with a thickness of about 50 m. The salinity anomalies in this layer are typically 0.02 psu while the temperature is higher by about 0.05 $^{\circ}$C at the deepest three stations. The salinity anomaly dominates the density resulting in enhanced potential density. This bottom layer is likely associated with the of bowl shape of the central plateau and we surmise that this is water that has become trapped

in the depression. This appears clear for the deepest part below 1700 m, but is not so obvious for the shallower signals at stations 134 and 138-140 which lie at about 1500 m and are thus well above the sill depth of the local depression. The properties near the bottom are similar to Makarov Basin properties, but the salinity is higher than the reference profile. The salinity and density match the Makarov Basin profiles 400 m deeper for the deepest stations along the section and about 200 m

deeper for the more shallow stations. The bottom layer has also anomalous concentrations of chemical constituents with significantly higher silicate concentration and lower oxygen concentration near the bottom (Fig. 5). These values also match the Makarov Basin properties 200-400 m deeper down. This suggests that Makarov Basin water has been uplifted 200-400 meter and filled up a bottom layer with relatively deep Makarov Basin characteristics across the high plateau of the passage. The salinity data

from a nearby moored instrument at 1700 m depth during 1995-1996 (Woodgate et al., 2001) show relatively large salinity variability, both short and long term, of about 0.02 psu which likely is





associated with vertical motions in combination with a vertical property gradient. The bottom-layer water characteristics can thus be explained by upwelling of water from deeper levels in the Makarov Basin that subsequently have become trapped on the plateau. The small vertical extent and sharp halocline at top of the layer is still puzzling, however. It indicates that the structure was formed

relatively recently since vertical turbulent diffusion would smear it out relatively quickly. The layer is also located at more shallow depth at the northern and southern stations which indicates that it is takes part in the across ridge flows giving rise to sloping isopycnals.

To provide additional evidence of water mass transport from the Makarov to the Amundsen Basin at the northern end of the southern gap we use two zonal sections at 81°N taken by RV Polarstern in

1995 and 1996 that coincide with the northern end of the SWERUS-C3 section. These sections include deep stations in both basins that can be used as reference stations to account for the inter-annual variability and long-term trends that have been identified in the Arctic (Polyakov et al., 2012). Both the 1995 and 1996 Atlantic Water layers are around 0.03 psu fresher, but have a similar core temperature as SWERUS-C3 data from 2014 (Fig. 6 c, d, e, f). In 1995, conditions show remarkably

little difference between 400-1000 m depth west and east of the ridge with a salinity minimum between 500 and 700 m (Fig. 6 c and g), originating from the Barents Sea branch inflow (Schauer et al., 2001). Deep water properties below 1500 m are similar to those observed during SWERUS-C3, so that we can use the latter to identify the origin of these water masses. In the 1995 section, the three stations furthest west (Fig. 6 g, blue) and east (Fig. 6 g, red) show clear Amundsen and Makarov deep

water characteristics, respectively. Two stations west of the ridge around 138°E (Fig. 6, a, c and g, light blue) show Amundsen Basin characteristics down to 1700 m and saltier Makarov characteristics below 1700 m. The deepest part includes a thin bottom layer of fresher, but colder Amundsen water. A station taken in the same position in 1996 (Fig. 6, b, d and h, magenta) shows a similar behavior. This suggests that the thin, salty bottom layer we observed in the SWERUS-C3 data is not an isolated

phenomenon in time and that the Makarov deep water that has been uplifted onto the ridge may spread westwards across the rugged western edge of the bowl in the southern passage. In 1996, all but the westernmost station along 81° N (Fig. 6, b, d and h, blue), show fresher Makarov water in the depth interval 500-1500 m, as referenced to the two easternmost stations (Fig. 6, b, d and h, red). This confirms our conclusion from the SWERUS-C3 data that there is indeed a westward transport of water

at the northern end of the southern passage in this depth interval.

Both the SWERUS data and historical sections thus show signals of a flow of Makarov Basin water towards the Amundsen Basin crossing the LR through the southern passage. This type of flow was also inferred from mooring data in 1995-1996 showing clear signals of Makarov Basin water at a mooring site (located at 81 34.5 N, 138 54.0 E) at the western flank of the LR and just north of the

southern passage (Woodgate et al., 2001).



### 3.2 Northern passage

The northern section of the LR shown in Figure 2b was the last area to be investigated in detail during the SWERUS-C3 2014 expedition. The limited time available before IB *Oden* had to begin the return transit implied that all oceanographic stations were carried out concurrently with the multibeam
mapping program. This excluded the possibility to strategically place hydrographic stations along section lines based on a detailed multibeam map of the area.

The multibeam mapping reveals a trough-like structure in the LR extending from about 84°24'N - 85°3'N in north-south direction and 148°E -151°E in east-west (Fig. 2b). The approximately 72 km long and 33 km wide trough forms an intra basin in the LR morphology, which not is well expressed
in IBCAO Version 3.0. The name *Oden Trough* for this feature has been formally accepted by GEBCO's (General Bathymetric Chart of the Oceans) Sub-Committee for Undersea Feature Names (SCUFN) in 2015. The eastern side of Oden Trough is enclosed by rather steep walls rising 500-600 m above a generally 1600 m deep trough bottom. The deepest passage, i.e. the sill, between the Makarov and Amundsen basins is 1470 m and located in the eastern wall at about 84°43'N 151°15'E (Fig. 2b,
e). The western opening towards Amundsen Basin is located at the northern end of the trough and is deeper as well as substantially wider than the eastern sill. The deepest part of Oden Trough is about 1704 m deep and located just northwest of the sill along the foot of the eastern wall, close to hydrographic station 152 (Fig. 2b). This deepening extends ~10 km along the foot of the steep wall north of the sill and is well pronounced in the bottom morphology.

The TS profiles in this area (Fig. 7) show a quite irregular behavior with influences of water mass properties from each side of the ridge at different depth intervals. Station 153, located in the Makarov Basin, closely follows the reference profile in the Makarov Basin below 1400 m, but in the depth interval between 1400 – 1100 m it has more Amundsen Basin properties. At 1100 m there is a sharp transition towards more Makarov Basin properties up to 800 m. Further up the profile at 700 m depth
there is again a transition towards Amundsen Basin properties. In the Atlantic layer (300-600 m) the water mass has Makarov Basin properties as demonstrated by the salinity and temperature maximum at $\sigma_\theta = 27.94$ kgm$^{-3}$. The nearby station 152 on the Amundsen Basin side of the ridge shows more Amundsen Basin characteristics below the 1470 m sill depth as indicated by the salinity profile but is similar to the Makarov Basin in the TS plot. Further up in the column it follows the Amundsen Basin
properties relatively closely, up to about 800 m, where it joins the Makarov Basin profile and has Makarov Basin properties all the way up to 300 m.

The station located at the western end of the of the passage close to the Amundsen Basin (stn. 155) has a structure more like the Amundsen Basin characteristics in the deepest part below 1400 m. In the interval 900-1400 m it has a mixture of properties. Above 900 m it undulates between clear Makarov





and Amundsen Basin characteristics up to 700 m and Makarov properties dominates above this depth. Station 154 is located in a more central part of the trough where there is a local depression. This station also undulates between different water mass properties but is clearly more dominated by Amundsen Basin characteristics especially in the interval 400-800 m where it is has significantly
higher salinity.

It is difficult to interpret this complicated water mass structure based on a few semi-synoptic CTD stations but it is possible to deduce some general aspects. It appears that this is an area of active water exchange across the LR presenting properties from the basins on either side. The profiles imply that this exchange is not an organized flow in one direction but more likely interleaving motions in
relatively distinct layers that can go in both directions and are probably intermittent in nature. An indication of the horizontal spatial structure of these flows is that Makarov Basin properties appear to dominate for stations 152 and 155 in the depth interval 300-900 m. Both of these stations are located near the northern high plateau suggesting a general flow from the Makarov Basin along the steep slope, but most of this flow appears to be occurring above 1000 m. On the other hand we see
Amundsen Basin characteristic's around 700 m in the profile on the Makarov Basin side of the ridge indicating a flow from the Amundsen Basin.  Additionally, the profiles from the Amundsen Basin side of the ridge show Makarov Basin properties below the sill depth indicating spillover of denser water from the Makarov Basin that is stored in the flat central area.

A 2011 Polarstern section (Schauer et al., 2012) crosses the LR at 84°30'N and provides a useful
complement to the SWERUS-C3 stations. It shows an AW water core that is cooler and fresher than in 2014 (Fig. 8). The two westernmost (Fig. 8, dark blue) and easternmost (Fig. 8, red) stations show coherent water mass properties of the Amundsen and Makarov Basins, respectively and we will use them as reference stations for the two basins. Similarly to the SWERUS-C3 data, the 4 stations over the ridge show evidence of water mass exchange across the ridge. A station at 140°E (Fig. 8, light
blue), west of the ridge, has the coolest and narrowest AW layer of the section with salinities in the 300-700 m depth range similar to those on the Makarov side, but follows Amundsen characteristics below 800 m. A station east of the ridge at 156°E (Fig. 8, magenta) has Makarov AW and deep water characteristics, but shows signs of an intrusion of water from the Amundsen side and interleaving between 600 and 1200 m. Two stations at 145°E (Fig. 8, green) and 150°E (Fig. 8, yellow) show
further indication of water mass exchange across the ridge. Notable is that the more western (Fig. 8, green) of those stations has the cold and fresh Makarov AW characteristics down to 1300 m and follows the Amundsen deep water below.

Both the SWERUS and historical data show that the northern passage is an area of active water exchange across the LR. It is indicated from the profiles that this exchange occurs as midwater
intrusions bringing water properties from each side of the ridge in well-defined but irregular layers.





Similar irregular intrusions are found at the Kara Sea slope east of the St. Anna Trough where the colder and fresher Barents Sea branch of the Atlantic inflow enters the Arctic Ocean and meets the warmer and saltier Fram Strait branch (Rudels et al., 2000; Rudels et al., 2013). A narrow front is formed in the confluence zone which is highly unstable for horizontal perturbations due to double

diffusive fluxes when colder and fresher water comes on top of a layer with warmer and saltier water (and vice versa). The perturbations then grow to large amplitudes and form the irregular layers. A similar situation, with a narrow front between water masses of different TS properties and near equal density, seems to be the case at the eastern wall of the Oden Through where water with Amundsen Basin properties comes in close contact with water having Makarov Basin properties. Since the LR has

a strong steering effect with flows generally along the ridge it reduces the water exchange between the major ocean basins and maintains a frontal structure between the water mass properties on each side which is typically seen on hydrographic sections crossing the ridge (Rudels et al., 2013). Note that the flows on each side go in the opposite directions and therefore typically bring different water mass properties. At locations where the ridge morphology is narrow it can be expected that this front will be

sharper. The narrow eastern wall of the northern passage (including the sill) appears to be such a feature which can guide the water masses streaming at each side of the ridge to come close together, sharpening the horizontal property gradient and form irregular intrusion. It can be expected that these intrusions should have a very small across frontal speed since they are driven by the minute density gradients generated by the double diffusive fluxes (McDougall, 1985).

On the other hand there are morphological expressions in the ridge seafloor that may result from more energetic flows in the vicinity of the sill. There is a nearly 100 m deepening, beginning at the sill to continue northward along the northern footwall of Oden Trough, possibly formed by long-term seabed erosion of bottom currents (Fig. 9). The sub-bottom chirp sonar data collected along with the multibeam bathymetry show disturbed bottom sediments closest to northern footwall as well as

transparent lenses with an acoustic appearance and stratigraphic location suggestion that they may be comprised of sediments redistributed by bottom currents (Fig. 9).

## 4 Discussion

We finally discuss some general aspects on how the LR effects the water circulation and property distribution in the Arctic Ocean. The deepest waters are directly isolated from each other by the ridge

and there is a clear contrast in properties between the opposite sides with deep waters in the Amerasian Basin warmer and saltier water than those in the Eurasian Basin. As noted above the ridge also affects the property distribution further up in the water column as seen in many hydrographic sections crossing the ridge (Rudels et al., 2012). The main reason for this property contrast is that the flow is largely barotrophic with small vertical shear and tends to follow the bathymetry even far away

from the bottom. It is therefore difficult for the flow to pass the ridge except at places where isobaths





cross the ridge. The effect of the ridge is much less in the upper layers above 100 m, including the halocline and the surface mixed layer, where the circulation is more controlled by wind forcing and strong horizontal density gradients associated with the supply and spreading of freshwater from rivers and inflowing low salinity water through Bering Strait (Morison et al., 2012). The horizontal density

gradients in the upper layers generate baroclinic motions which are decoupled from the topographical steering.

The southern passage provides a direct pathway above 1700 m for the warm Atlantic water streaming along the continental slope to enter the Amerasian Basin. It is mostly water from the colder Barents Sea branch that passes the ridge while the warmer Fram Strait branch water turns and follows the ridge

towards north (Rudels et al., 2013). A substantial part of the Fram Strait branch can also turn earlier and follow the Gakkel Ridge back towards the Fram Strait (Rudels et al., 2013). This means that most of the hydrographic sections crossing the ridge show a warmer Atlantic water core on the Amundsen Basin side than on the Makarov Basin side. The dynamics that actually determine the properties of the water passing the ridge, of which the temperature is a critical quantity, appears to be an open question.

The temperature of the Atlantic water including the seabed temperatures the entire Amerasian Basin should be highly dependent on this control.  Current meter data from three moorings located at the Siberian shelf slope (one at each side of the L R) and one at the L R slope (Woodgate et al., 2001) show that the mean velocities are small and follow the depth contours but there are also numerous and strong cross-isobath flow events which are mostly related to mesoscale eddies but also likely due to

wind forced up and down-welling. This means that not only topographic steering, but also other types of dynamics will control the passage of water masses across the ridge. Also the mixing between the Barents Sea and Fram Strait branches along the continental slope upstream from the LR will be critical.

The southern passage appears also to guide a westward flow of Makarov Basin water towards the

Amundsen Basin as seen here and in earlier investigations. This flow is likely of intermittent nature which is evident from the 1700 m mooring data in the Amundsen Basin side where the TS properties undulate between Makarov and Amundsen basin signatures (Woodgate et al., 2001). Our section shows that this flow is present over a quite large depth range from the bottom and up to at least 500 m depth.

The northern passage seems to provide a different type of exchange in terms of interleaving motions above the1470 meters sill depth, with flow in opposite direction over different depth intervals. It is likely that this flow structure is less important for the exchange of properties across the ridge since velocities associated with double diffusive interleavings should be small ~5 mm s$^{-1}$ (McDougall, 1985). Combining this with the relatively small horizontal scale of the passage of ~50 km and a

vertical scale   ~100 m results in a rather small volume flow ~0.05 Sv compared with the 2 Sv



estimated to pass eastward through the southern channel (Woodgate et al., 2001). However, the interleaving signals of Makarov Basin water are seen at quite large distances from the sill which means that they actually carry properties across the entire LR and should therefore contribute to the overall water exchange across the ridge, but the question is to what extent.  It is hard to say anything

more specific from this type of data.

There are also other deep channels which are pathways for flows crossing the ridge. Earlier investigations show that Canadian Basin Deep Water crosses the ridge through the 1870 m deep passage (the deepest passage) near the North Pole (Björk et al., 2007) and can be traced along the Amundsen Basin slope towards Greenland and further along the northern Greenland continental shelf

slope (Björk et al., 2010). Signs of this water are also seen as a salinity maximum around 2000 m over a large part of the Amundsen Basin. Closer to Greenland where the ridge meets the continental slope there is a passage with sill depth of around 1200 m. The overall bathymetrical structure in this area is similar to the passage near the Siberian continental slope at the other end of the ridge. This will cause a bifurcation of the westward rim current along the northern Canada continental slope, as also noted

by Rudels et al. (2000). The shallow part crosses the ridge while the deeper part turns northward and flow towards Siberia along the LR flank on the Makarov Basin side. There are also indications that a part of the northward flowing branch is guided to cross the ridge hugging the slope around a 500 m depth shoal or at some deeper passages between the southern 1200 m passage near Greenland and the deepest 1870 m passage (see Björk et al., 2010 for details).

In a larger scale context it is clear that the LR not only plays a critical role by shaping the property distribution of the intermediate and deep waters among the major Arctic Ocean basins but also by controlling the mixing and water mass transformation from the inflowing Atlantic water contained in the two branches to the outflow of colder and fresher modified water through the western Fram Strait (Rudels et al., 1999). This outflow then contributes to the overflow across the Greenland-Scotland

ridge and the north Atlantic deep water formation. It should therefore be valuable to make further and more detailed studies of the different deep passages in the LR including moored instruments to obtain time series.

**Acknowledgments**

The SWERUS-C3 expedition was financed by Knut and Alice Wallenberg Foundation,

Swedish Polar Research Secretariat and Stockholm University.  Research grants to Larry Mayer was provided by the U.S. National Science Foundation Grant No. PLR-1417789. We also thank the crew of *I/B Oden* for all their professional help during the cruise. The 1995 and 1996 Polarstern data were received via the National Oceanographic Data Center.



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



**Figures**

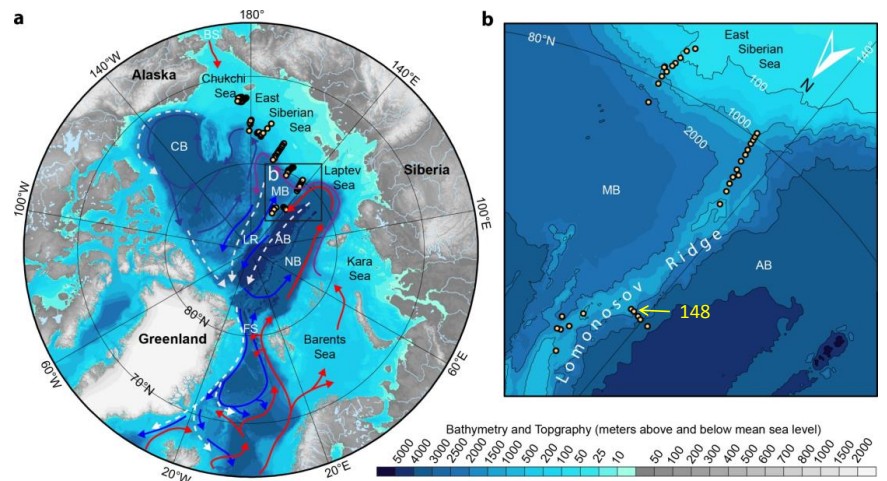

**Figure 1:** a) Map of the Arctic Ocean with the major ocean currents. Red arrows: flow of warm
Atlantic water and inflow Pacific water through Bering Strait. Purple: flow of colder Barents Sea
Branch of the Atlantic water inflow which has been modified when crossing the Barents Sea. Blue:
flow of colder and fresher modified Atlantic water. The red and blue arrows also represent the deep
water circulation below the warm core of Atlantic water. White hatched arrows: Flow of low salinity
water in the surface layer and halocline. Abbreviations: Lomonosov Ridge (LR), Fram Strait (FS),
Bering Strait (BS), Nansen Basin (NB), Amundsen Basin (AB), Makarov Basin (MB), and Canada
Basin (CB). Yellow dots show the positions for all CTD station during the SWERUS-C3 expedition.
b) Details of the special study area of the LR including the CTD stations. Note the position of station
148 which is referred in the text. For further information of station numbers see Figure 2.

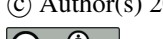



**Figure 2:** Detailed maps of the southern passage (a, c) and northern passage (b, e). The likely position of the sill for the southern passage is within the white circle. A semitransparent plane visualize the approximate sill level of 1700 m for the southern passage (c) and the actual sill level of 1470 m for the northern passage (e).





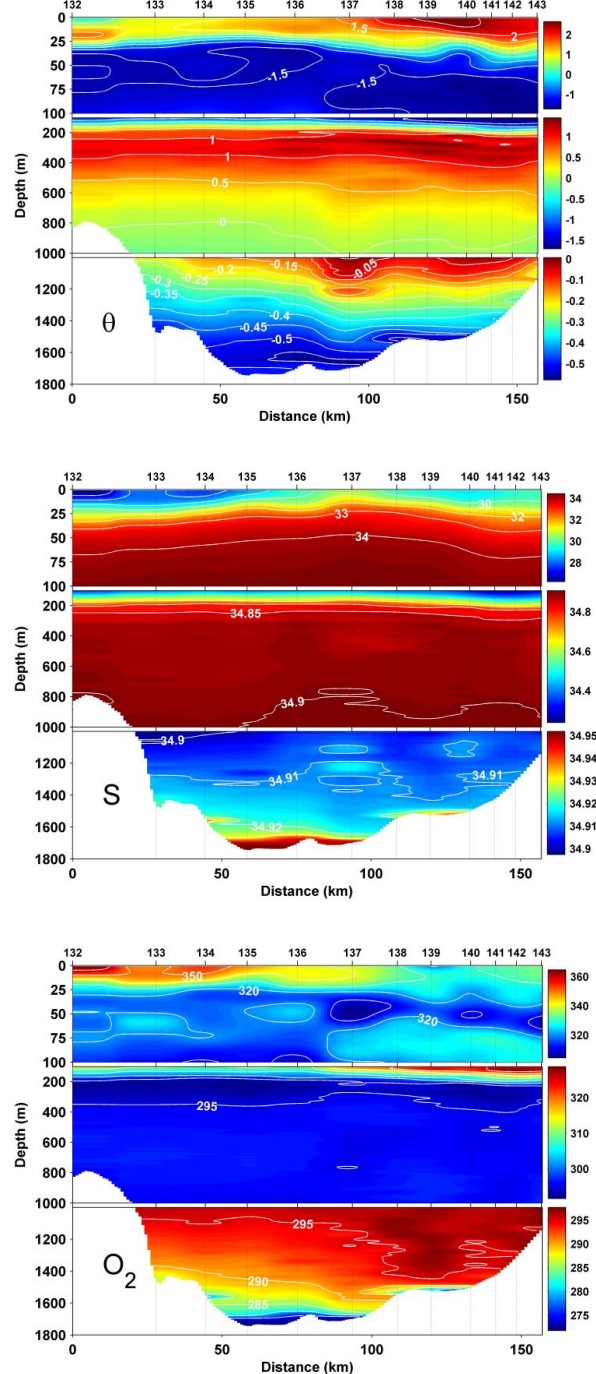

**Figure 3:** Potential temperature θ ($^o$C), salinity S (psu) and dissolved oxygen $O_2$ (μmol kg$^{-1}$) sections
5       across the southern passage.





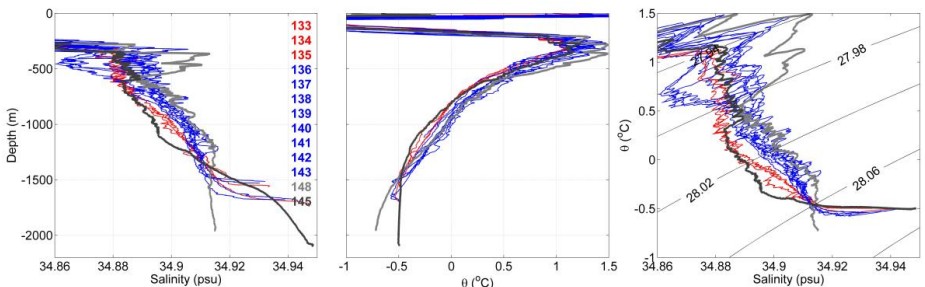

**Figure 4:** Salinity and potential temperature (Θ) profiles in the southern passage. The bold profiles show reference stations in the Makarov Basin (Stn 145, black) and Amundsen Basin (Stn 148, gray). Blue profiles are those at the southern end of the section, the three northernmost are red. For exact locations of the stations, please refer to Fig. 1.





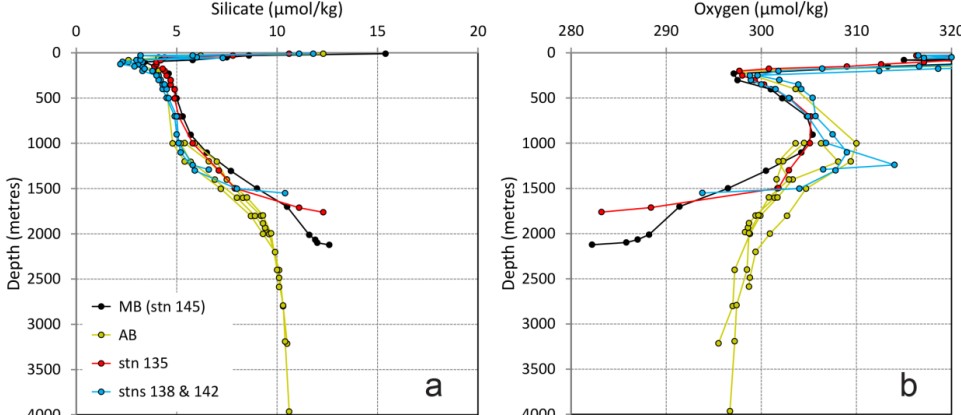

**Figure 5:** Silicate (a) and dissolved oxygen (b) in the southern passage.





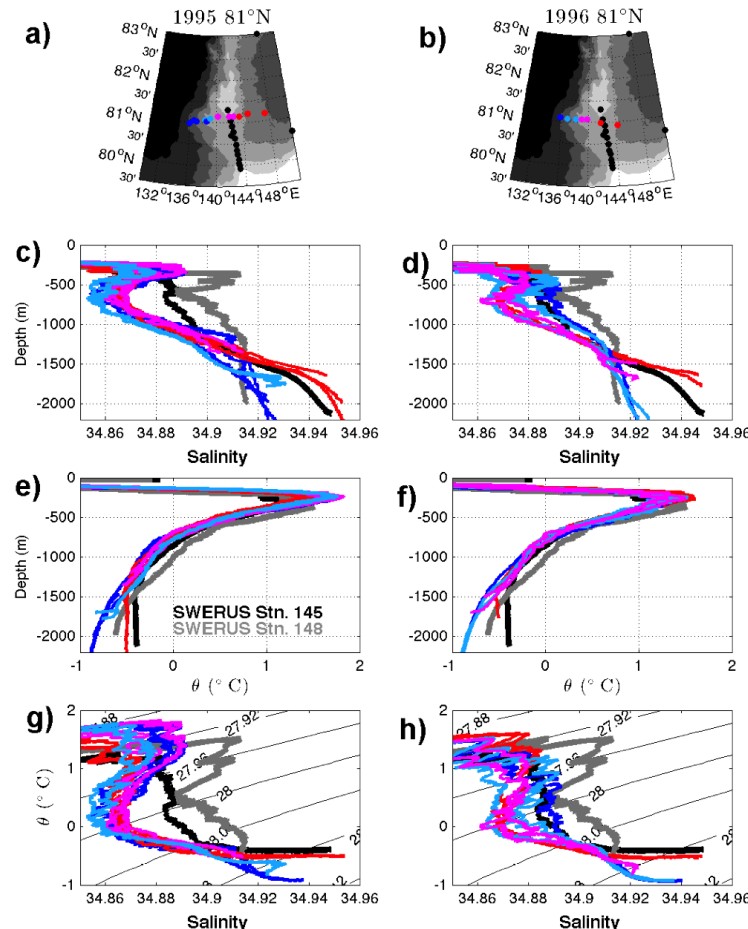

**Figure 6:** Historical water mass distributions across the Lomonossov Ridge. Potential temperature and salinity profiles and potential temperature-salinity diagrams for 2 sections taken at 81°N in 1995 (a, c, e, g) and 1996 (b, d, f, h). The black and gray profiles mark the SWERUS reference stations for the
5  Makarov (station 145) and Amundsen (station 148) basins, respectively. The historical stations are colour-coded according to longitude and water mass characteristics and are marked in the same colour in the maps in panels a and b.





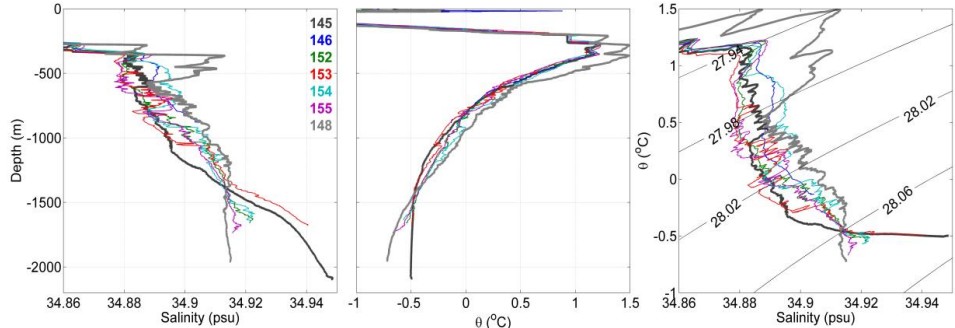

**Figure 7:** Salinity and potential temperature (Θ) profiles at the northern passage. The bold curves show reference stations in the Makarov Basin (Stn 145) and Amundsen Basin (Stn 148).




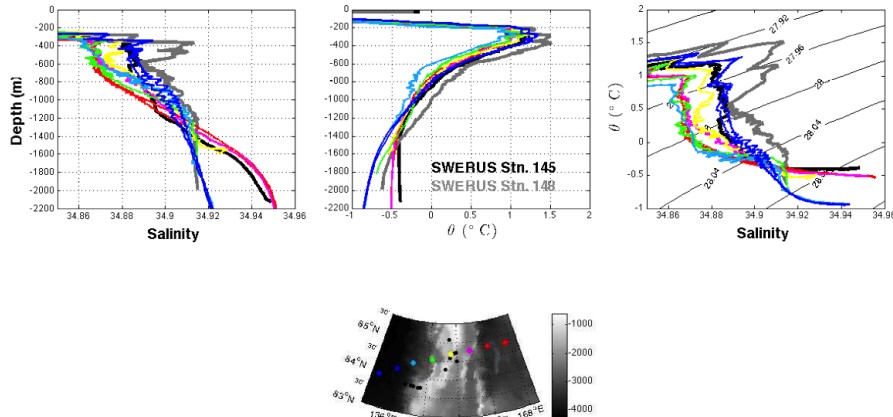

**Figure 8:** Water mass characteristics across Lomonosov Ridge at the northern passage from a 2011 Polarstern cruise. The black and gray profiles mark the SWERUS reference stations for the Makarov (station 145) and Amundsen basins (station 148), respectively. The historical stations are colour-coded according to longitude and water mass characteristics. Please refer to the map for their exact position.





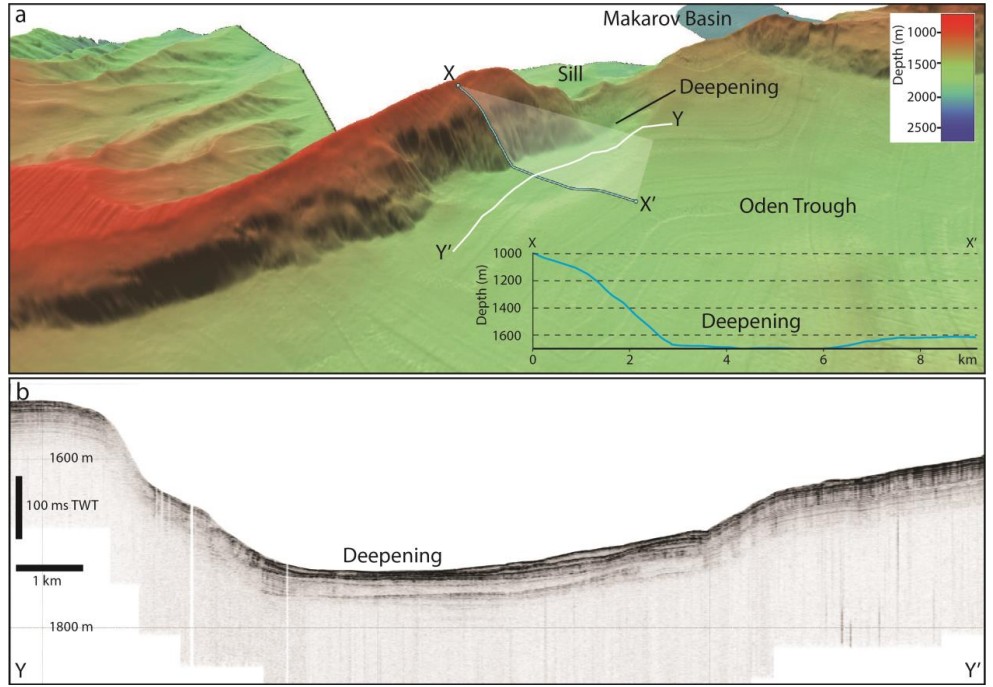

**Figure 9:** a) Detail of the northern passage in the vicinity of the sill showing a deepening which is possibly formed by long term seabed erosion of bottom currents. b) Sub-bottom chirp sonar data along the Y-Y' section.

