# Peer review of "Bathymetry and oceanic flow structure at two deep passages crossing the Lomonosov Ridge"

_Ocean Science, 2017_

## Referee Comment (RC1) · Anonymous Referee #1 · 26 May 2017

GENERAL COMMENTS

This paper presents and discusses recent bathymetric data and hydrographic features in the Arctic region of the Lomonosov Ridge, which is characterized by complex dynamics. If on the one side, the topic should deserve publication thanks to the detailed bathymetric survey and the collection of CTD casts in a very (scientifically) interesting region, on the other side I cannot recommend the publication of the manuscript (ms) in its present form. Unfortunately, the paper still suffers from a number of weaknesses and major revisions are required. While "Abstract" and "introduction" are pretty well organized and clear, "methods", "results", and "discussion" need strong improvements. I have given some general and (not all) specific comments below that I hope will help the Authors to prepare a more robust version of the paper in the next future. In particular, the description of the thermohaline properties needs to be re-organized. Finally, I

strongly recommend the Authors to review the English through a native English speaker before next submission.

SPECIFIC COMMENTS Hereafter, I will report some comments on each section of the ms.

Methods

This section misses many information: detailed period of the cruise (days, months), explanation about collection and use of silicate and dissolved oxygen data (maybe coming from discrete water samples and laboratory analyses?). Moreover, if you compare your 2014 data with previous ones, especially if the latter are taken 10 years before, you should justify your choice, being aware that the comparison of thermohaline properties taken in period much different can raise some criticisms. A minor comment: I suggest the Authors to write somewhere in the methods that they analyze thermohaline properties by using potential temperature, salinity and dissolved oxygen, and define those parameters and their symbols once and for all.

Results

In section 3.1 the Authors start describing the bathymetry of the region. I had to spend a time to compare figures 1 and 2: I would suggest the Authors to use the same criteria of orientation in panels of figure 2 (a, b) with respect to that in figure 1 (b). It would render easier for the reader to visualize regions and possible pathways of water masses in the study area discussed later within the ms.

Page 5: After line 20, Authors describe the hydrological properties. I suggest adding a phrase that could connect the first part of the section 3.1 with its second part.

Line 21: substitute "the section across the passage" with something like "the north-south hydrological section across the passage. . ."

Lines 27-29: Are you speaking about the surface warm core? If so, please move up this phrase within the text, where you are speaking about the surface layer, otherwise

it is difficult to follow the description of the thermohaline properties.

Lines 31-32: explain why such properties of the bottom layer are "anomalous". From figure 3, which this part of the text is referred to, the mentioned temperature increase at the very thin bottom layer is not visible. Moreover, cold and salt waters, hence very dense (why you do not add also potential density data?), are normally trapped in the bottom layer of a basin, and lower dissolved oxygen values confirm that they are also pretty old (i.e. not ventilated since a relatively long time).

Line 34: remove double 'to be'.

Lines 35-37: please better explain this part and/or support with adequate references (e.g. Chelton et al 1997, JPO). How did you calculate the Rossby radius? Why did you not mentioned this calculation in the "methods"? Again, did you consider checking satellite images, sea level, or horizontal distribution of potential density to see if any eddies would be visible? It could support the discussion.

Page 6:

Lines 4-5: remove "matching the salinity and temperature data". Perhaps, you could discuss the origin of this bottom water masses, and how they accumulated in this part of the ridge, remaining likely isolated from the rest of the water column.

Lines 6-7: please change this part with "Vertical profiles of $\theta$ and S collected in the southern passage of the LR are compared with those collected in the Makarov Basin (Stn. 145 in fig. 2b) and Amundsen Basin (Stn. 148 in fig. ???) and shown in figure 4."

Lines 7-19: this part need to be re-written to render it clearer. I was totally lost reading this part. Please write in more orderly manner about layers, water masses, possible pathways, explain why you chose two reference stations, and so on.

Lines 20-36: the same comment as above: re-write this part to be more clear. Authors start introducing silicate data without any previous explanation of them in the "methods". Additionally, they suddenly refer to mooring data gathered in 1995-1996 but this

part is not well inserted in the context.

Page 7:

Line 2: "upwelling" is a specific oceanographic process, in this case is it driven by what? Maybe Authors mean "upward displacement"

Line 5: "turbulent diffusion"? ok, but please justify or refer to appropriate bibliography.

Lines 8- end of the page: to be re-organized. Again, comparing 2014 thermohaline conditions with those of 1995-1996, after you have written that the variability is large in this region, does not make any sense, unless you justify this approach. Moreover, remove conclusions from this section.

Page 8:

Lines 1-6: move this part to the "methods".

Lines 20...: the same comment used for the previous page. The description of the thermohaline properties distribution is confusing, and need heavy improvements, in terms of language used and organization of the text.

Discussion:

Based on the comments I have provided for the results, I could say that the discussion has to be revised accordingly to the future changes required for the "results". However, I will provide here some comments: Page 10, Lines 34-35: how do you define the flow "largely barotropic"? Is it reported in literature?

Page 11, line 7: I do not think that "water streaming" is appropriate, please check it.

Page 11, Line 11: indicate "Gakkel Ridge" in figure 1.

In general, it seems to me that parts of the discussion could be moved to the introduction, while here the Authors should discuss their own data with more detail. Doing so, they could provide some nice conclusions (now they are not clear) on water masses

distribution (as they did) and some speculations or hypotheses on the evolution of the thermohaline properties according with previous already published data.

FIGURES:

Figure 3: To use different color scales in each layer can be useful to see the variability within each of them, but can confuse the reader because it seems that (e.g.) intermediate and deep layers have different values while they are almost similar with the exception of the very bottom layer. Try to use the same scale in each layer.

Figure 4: respect always the same order, $\theta$ first, S second, Dissolved oxygen third (if you want to show), and then $\theta$/S diagram. In general, figure is not clear, all profiles seem bold, and colors between st. 145 and 148 are not clearly distinguishable. Finally, for the exact location of the stations, it is better to refer to figure 2, not figure 1.

Figure 5: from this figure, it seems that silicate and dissolved oxygen data comes from discrete water samples. Why the Authors did not described this aspect in the "methods"?

[Figure]

---

## Referee Comment (RC2) · Anonymous Referee #2 · 3 Jul 2017

General comments: This paper presents new oceanographic and multibeam bathymetric data over a little known part of the Lomonosov Ridge and discusses the exchange of water masses across two channels in the ridge. The data presented and conclusions reached seem to be sound with good comparisons with historical oceanographic data from the area. Overall, I find the discussion a little lacking as it mostly reviews existing data rather than placing the data presented in this study in context and discussing it more fully. What are the broader implications of the new study? For example, on seabed temperatures and the GHSZ, as raised in the Introduction. Or on heat exchange with the Arctic Ocean? Can the authors comment on any unresolved or new questions raised by the study? I found the paper to have a clear structure and the figures are sufficient to provide evidence for the results and discussion sections.

[Figure]

However, some parts of the text need revising to improve the written English/clarity of the sentences and likewise the figures can be made more consistent and clearer with some small edits (see specific/technical comments). If the authors can address the below comments, in particular developing the Discussion and relevance of the study, and revising the figures and written text appropriately then I can recommend publication in OS.

Specific comments: P6, Line 11: Could you colour stations 137 and 138 differently so that this statement is clearer on Fig 4? I can just about see what you mean, lower salinities c 500 m, but all profiles being blue it is hard to distinguish from the other profiles for the remaining depths

P7, line 14 & figures: Please make all multi-panel figures in the paper a,b,c etc. You have done this for some figs but not all. Make consistent across the figures, and also how you label them and the fonts used (some bold with a bracket, some not bold no bracket…). You can then refer more easily to the salinity/potential temperature plots in the text.

P7, line 27: Any comment on this westernmost station? Marakov water just not reached here or Amundsen signal overwhelming, barrier to W transport/mixing?

P9 line 9: Interleaving motions = or could it by a gyre/circulation within the intra basin? Perhaps explain the origin of "interleaving motions" if this is the correct oceanographic term (not my speciality)

P10, line 25: Label disturbed bottom sediments and transparent lenses on Fig 9

P12 line 4-5: I would prefer to see a comment on what kind of data could be used to elucidate the flow exchange rather than a negative comment on the data presented in this study! Otherwise why are we publishing it?

P12 line 20-27: The discussion lacks discussion on larger implications of this work plus any comments on unresolved/new questions raised by the study. What is the

implication, if any, on seabed temp, for example.

Technical corrections: P5, Line 15: degrees symbol missing on lat/long. Check the rest of the text for this as I also noted it elsewhere (e.g. p7).

P6, Line 15: sentence needs re-writing/punctuation as it is a little confusing, e.g. . . .lower concentrations in the northern deep part of the section. . .

P6, Lines 16-19: Which observations, up to now you are talking about the three northern stations. Check and clarify the last sentence. Change to "Therefore, in addition to flow from the Makarov Basin in the north, there is clear evidence for flow from the Amundsen to the Makarov Basin in the depth interval 400-1300m"?

P7, line 6: Remove "is"; doesn't quite make sense, re-word slightly: ...indicates that it contributes to cross-ridge flows giving rise to sloping isopyncnals

P7, line13: Is Atlantic Water always with a capital W? Check here and throughout the text, I thought the standard was for a small w in the literature. . .

P8, line 5-6: Written English not great here, suggest changing it to: ". . .excluded the possibility of strategically placing hydrographic stations along section lines based on the detailed multibeam map. . ." I'd actually replace detailed bathymetry map, rather than multibeam map, although the data was collected by a MBES what you are actually mapping is the bathymetry. . .

P8, line 8: I think this should be 72-km long and 33-km wide; could check OS hyphenation policy

P8 line 10: Replace has been with was. Refer to Fig 2e

P8 line 12: Remove acronym SCUFN as not used anywhere else in the text

P8 line 17: Is "about 1704 m deep"?! 1704 m is pretty specific! I'd suggest removing about

P9 line 19: Refer to Fig 8d after latitude

P9 line 25: Have you defined AW acronym? If so then please use throughout the text; check

P9 line 27: What is Makarov AW? I know it has been discussed but perhaps define it by its T/S properties in brackets if you are going to talk about this as a distinct water mass

P9 line 34: Passive voice not great in this sentence; consider replacing "It is indicated from the profiles that this" with "The TS profiles indicate that this"

P10 line 4: Replace for with in terms of

P10 line 6:"grow to large amplitiudes" not great written English. Replace with "..grow in amplitude and form.."?

P10 line 8: Should be Oden Trough

P10 line 9: Replace in with into. I'd replace having with with

P10 line10: Add punctuation: "Since the LR has a strong steering effect, with flow generally along the ridge, it reduces the water…"

P10 line 16: Replace at each with on either; remove close

P10 line 17: and forming irregular intrusions. Replace can be with is

P10 line 18: Remove should; frontal speeds

P10 line 21: …deepening, from the sill northward along the northern footwall of Oden Trough,…

P10 line 31: Comma after above

P11 line 15: Word missing here? Seabed temperature in the entire…? Commas after water and Basin

P11 line 16-20: Long sentence! Consider splitting into two

P11 line 21: Remove will; comma after Also

P11 line 28: Replace bottom with "seabed upwards,"

P11 line 31: Should be directions

P12 line 10: water mass?

P12 line 12: bathymetrical structure…yikes! Overly complicated I think, suggest change to bathymetry or seafloor morphology. Structure implies tectonic influence to me, or internal Earth processes – bathymetry or morphology seems appropriate in this case

P12 line 25: I thought NADW was a widely used acronym, capitalise North Atlantic Deep Water

P12 line 25-27: Weak last sentence which needs rewriting but see my more general comments on developing the Discussion of this paper so I hope that this sentence will be revised

Figures, general: Please make sure fonts and labels for each panel are the same in each figure, and of an appropriate size to be read. Label panels in Figs 3, 4, 7, 8 as a, b, c etc and then refer to specific panels in the text.

Figure 6: The lines on c-h are to thick to distinguish in many places. At least at the scale that the figures are reproduced in the PDF. Check that these can be clearly seen, Figure 7 is much better so please make consistent.

Figures 4, 6, 7, 8: Cannot have negative depths..! And you do not have this in your other figures, please remove negative signs in front of depths

Figures 6, 8: Maps are very small, at least in the PDF version, and labels are too small and not clear. Revise.

[Figure]

---

## Author Comment (AC1) · 1 Oct 2017

We thank the reviewer for a thorough reading of our manuscript and many insightful and constructive suggestions for revisions.

Summary
Referee #1 finds that the detailed bathymetric survey and collection of CTD casts presented in our paper from the Lomonosov Ridge are scientifically interesting and deserves publication, although first after a major revision. It is specifically the "methods", "results" and "discussion" sections Referee #1 comments on. We have decided to undertake the revision. Referee #1 provided some specific comments that we will address one by one in our revision (The comments by Referee #1 are written in italics with blue text).

**Methods**
Missing information: *Detailed period of the cruise (days, months), explanation about collection and use of silicate and dissolved oxygen data (maybe coming from discrete water samples and laboratory analyses?)*

This information will be added in our revision.

*Moreover, if you compare your 2014 data with previous ones, especially if the latter are taken 10 years before, you should justify your choice, being aware that the comparison of thermohaline properties taken in period much different can raise some criticisms.*

The choice for our data comparison is simple, it is it the only data we possess from this very sparsely investigated area of the Arctic Ocean. We strongly believe that a comparison with the older data collected from previous years is important to show since they overlap spatially and support the notion that our observations not only represent anomalous snapshots. However, we agree with Referee #1 in that it is important to emphasize the time differences between the data sets. We will therefore add some clarifying words in the revised paper, and include more clearly from when over the year all data shown in the paper was collected, and not only at what year.

The following will be included:
*"Additional evidence of water mass transport from the Makarov to the Amundsen Basin at the northern end of the southern gap is found in two zonal sections at 81°N acquired by RV Polarstern expeditions ARK-XI/1 and ARK-XII in 1995 and 1996 respectively. The 1995 section was acquired August 20-27 and the 1996 section August 16-20. They coincide with the northern end of the SWERUS-C3 section and include deep stations in both basins. This implies that they may serve as reference stations representative for each year to exclude inter-annual variability and long-term trends that have been identified in the Arctic (Polyakov et al., 2012)..*

*A minor comment: I suggest the Authors to write somewhere in the methods that they analyze thermohaline properties by using potential temperature, salinity and dissolved oxygen, and define those parameters and their symbols once and for all.*

This will be added as suggested.

Results

*In section 3.1 the Authors start describing the bathymetry of the region. I had to spend a time to compare figures 1 and 2: I would suggest the Authors to use the same criteria of orientation in panels of figure 2 (a, b) with respect to that in figure 1 (b). It would render easier for the reader to visualize regions and possible pathways of water masses in the study area discussed later within the ms.*

Point taken. We will rotate the main map so 140° is down, implying that it fits with the insets in later figures.

[Figure]

Top: Original map, Bottom: Revised rotated map.

*Page 5: After line 20, Authors describe the hydrological properties. I suggest adding a phrase that could connect the first part of the section 3.1 with its second part.*

We suggest bridging the two sections with:
*"Overall, the new multibeam bathymetry from SWERUS-C3 together with depth information from IBCAO 3.0 provides a spatial context for the north-south hydrographic section across the southern passage (along the ridge crest) close to the Siberian continental slope (Fig. 3)."*

*Line 21: substitute "the section across the passage" with something like "the northsouth hydrological section across the passage. . ."*

This will will be inferred in the bridging suggestion above.

*Lines 27-29: Are you speaking about the surface warm core? If so, please move up this phrase within the text, where you are speaking about the surface layer, otherwise it is difficult to follow the description of the thermohaline properties.*
No, this is referring to the Atlantic water, which is subsurface. We have removed "core" in the first sentence to avoid any confusion.

*Lines 31-32: explain why such properties of the bottom layer are "anomalous". From figure 3, which this part of the text is referred to, the mentioned temperature increase at the very thin bottom layer is not visible. Moreover, cold and salt waters, hence very dense (why you do not add also potential density data?), are normally trapped in the bottom layer of a basin, and lower dissolved oxygen values confirm that they are also pretty old (i.e. not ventilated since a relatively long time).*

Here, "anomalous" simply refers to a slight change in near bottom layer water properties; we will use more clarifying wording. It should be noted that the changes we refer to are also seen in Figure 4, specifically salinity. For this reason, the reader pointed to Figure 4 in this section in addition to Figure 3.
We will also show potential density data. Yes, normally oxygen content reflects age. Here also differences in source waters contribute to the oxygen contrast.

*Line 34: remove double 'to be'.*

Will be fixed.

*Lines 35-37: please better explain this part and/or support with adequate references (e.g. Chelton et al 1997, JPO). How did you calculate the Rossby radius? Why did you not mentioned this calculation in the "methods"? Again, did you consider checking satellite images, sea level, or horizontal distribution of potential density to see if any eddies would be visible? It could support the discussion.*

After some further analyze of the special characteristics of station 137 and 138 we have arrived at the conclusion that we probably see some type of interleaving structure between the Barents Sea branch and Fram Strait branch waters which is advected along with the boundary current. We will adjust the text according to this and thus exclude the discussion about eddy and the associated Rossby radius.

*Lines 4-5: remove "matching the salinity and temperature data". Perhaps, you could discuss the origin of this bottom water masses, and how they accumulated in this part of the ridge, remaining likely isolated from the rest of the water column.*

"Matching the salinity and temp……." will be removed.

*Lines 6-7: please change this part with "Vertical profiles of θ and S collected in the southern passage of the LR are compared with those collected in the Makarov Basin (Stn. 145 in fig. 2b) and Amundsen Basin (Stn. 148 in fig. ???) and shown in figure 4."*

This section will be revised for clarity.

*Lines 7-19: this part need to be re-written to render it clearer. I was totally lost reading this part. Please write in more orderly manner about layers, water masses, possible pathways, explain why you chose two reference stations, and so on.*

This section will be revised for clarity.

*Lines 20-36: the same comment as above: re-write this part to be more clear. Authors start introducing silicate data without any previous explanation of them in the "methods". Additionally, they suddenly refer to mooring data gathered in 1995-1996 but this is not well inserted in the context.*

This section will be revised for clarity and the measurement procedure of silica in water samples will be described in the methods.

*Page 7: Line 2: "upwelling" is a specific oceanographic process, in this case is it driven by what? Maybe Authors mean "upward displacement"*

Yes, it should be upward displacement.

*Line 5: "turbulent diffusion"? ok, but please justify or refer to appropriate bibliography.*

This will be expanded on in a separate sentence with reference included:

"Using a value of $5 \cdot 10^{-5}$ $m^2 s^{-1}$ for the turbulent diffusion coefficient according to observations at the Lomonosov Ridge (Rainville and Winsor, 2008), it would take about one year to reduce the maximal vertical salinity gradient with 50%."

*Lines 8- end of the page: to be re-organized. Again, comparing 2014 thermohaline conditions with those of 1995-1996, after you have written that the variability is large in this region, does not make any sense, unless you justify this approach. Moreover, remove conclusions from this section.*

The approach is now justified and explained with more metadata, as described above.

*Page 8.*
*Lines 1-6: move this part to the "methods".*

This will be moved to the methods. ¨

*Lines 20. . .: the same comment used for the previous page. The description of the thermohaline properties distribution is confusing, and need heavy improvements, in terms of language used and organization of the text.*

This section will be revised for clarity.

*Based on the comments I have provided for the results, I could say that the discussion has to be revised accordingly to the future changes required for the "results".*
*However, I will provide here some comments: Page 10, Lines 34-35: how do you define the flow "largely barotropic"? Is it reported in literature?*

Yes, the batropic flow is reported in literature, which we will include in the revised version (Noest and Isachsen, 2003).

*Page 11, line 7: I do not think that "water streaming" is appropriate, please check it.*

Will be changed to "water flowing".

*Page 11, Line 11: indicate "Gakkel Ridge" in figure 1.*

To be changed in revised figure.

In general, it seems to me that parts of the discussion could be moved to the introduction, while here the Authors should discuss their own data with more detail. Doing so, they could provide some nice conclusions (now they are not clear) on water masses distribution (as they did) and some speculations or hypotheses on the evolution of the thermohaline properties according with previous already published data.

The discussion will be expanded considering the broader implications of our results.

FIGURES: Figure 3: To use different color scales in each layer can be useful to see the variability within each of them, but can confuse the reader because it seems that (e.g.) intermediate and deep layers have different values while they are almost similar with the exception of the very bottom layer. Try to use the same scale in each layer. Figure 4: respect always the same order, θ first, S second, Dissolved oxygen third (if you want to show), and then θ/S diagram. In general, figure is not clear, all profiles seem bold, and colors between st. 145 and 148 are not clearly distinguishable. Finally, for the exact location of the stations, it is better to refer to figure 2, not figure 1. Figure 5: from this figure, it seems that silicate and dissolved oxygen data comes from discrete water samples. Why the Authors did not described this aspect in the "methods"?

It is actually necessary with different color scales for each depth interval since it otherwise is impossible to see the variability. We will add a note in the figure text to alert the reader about the different scales. We are working on figure 4 in order to make it clearer and will refer to figure 2 for locations. The discrete water sampling data will be included in methods.
We thank the reviewer for a thorough reading of our manuscript and many insightful and constructive suggestions for revisions.

**General comments:**

We think that the reviewer points to important general improvements concerning language, clarity and readability of figures, and the benefits of extending discussions on the broader implications of our study; in particular pointing to potential new questions emerging from our work. We will revise the paper based on these overarching suggestions, as will be detailed below.

**Specific comments:**
*P6, Line 11: Could you colour stations 137 and 138 differently so that this statement is clearer on Fig 4? I can just about see what you mean, lower salinities c 500 m, but all profiles being blue it is hard to distinguish from the other profiles for the remaining depths.*

We will introduce different line types for these two stations (solid and dashed lines) to facilitate distinguishing between them.

*P7, line 14 & figures: Please make all multi-panel figures in the paper a,b,c etc. You have done this for some figs but not all. Make consistent across the figures, and also how you label them and the fonts used (some bold with a bracket, some not bold no bracket. . .). You can then refer more easily to the salinity/potential temperature plots in the text.*

We will edit the figures concerning panel labels and fonts as suggested.

*P7, line 27: Any comment on this westernmost station? Makarov water just not reached here or Amundsen signal overwhelming, barrier to W transport/mixing?*

Most likely, the Makarov Basin water in this depth range is deflected along the Ridge bathymetry (in the north-south direction; see Woodgate and Nøst and Isachsen in the reference list) slightly to the east of the westernmost station, where the topography becomes steeper. We will add the following comment in the revised ms.:
"The absence of a Makarov Basin hydrographic signature at the westernmost station indicates that the Makarov Basin waters are deflected along the steep topography on the western side of the Lomonosov Ridge."

*P9 line 9: Interleaving motions = or could it by a gyre/circulation within the intra basin? Perhaps explain the origin of "interleaving motions" if this is the correct oceanographic term (not my speciality)*

We will rewrite the sentence related to interleaving, providing a reference to Rudels et al. (1999), who in detail review interleaving processes in the Arctic Ocean. The revised sentence reads:
"The profiles suggest that this exchange is not a unidirectional organized flow, but more likely interleaving motions resulting from double-diffusive mixing process, which can create relatively distinct layers with variable flow directions that are intermittent in nature (Rudels et al., 1999)."

*P10, line 25: Label disturbed bottom sediments and transparent lenses on Fig 9*

We will do this.

*P12 line 4-5: I would prefer to see a comment on what kind of data could be used to elucidate the flow exchange rather than a negative comment on the data presented in this study! Otherwise why are we publishing it?*

We will replace the sentence "It is hard to say anything more specific from this type of data." with "Information on the flow speeds, ideally from moored current meters, would be needed to decide the significance of the cross-ridge exchanges of volume, heat, and salt in this northern passage."

*P12 line 20-27: The discussion lacks discussion on larger implications of this work plus any comments on unresolved/new questions raised by the study. What is the implication, if any, on seabed temp, for example.*

We will replace the last paragraph in the discussion section with a new one pointing to a few new open research questions (see examples below) which are highlighted by the present study.

1) Our study suggests that the exchange flows across the Lomonosov Ridge have rather different character in the southern and the northern passages. In the southern one, the flow has a more coherent structure, comprised by a few vertical layers with unidirectional velocities. In the northern passage, on the other hand, the flow is broken up into multiple shallow vertical layers with alternating velocity directions. Thus, our results highlight the question of what physical and bathymetrical features that control the nature of the exchange flow through the deep passages cutting across the Lomonosov Ridge.

2) Our survey of the bathymetry and hydrography in the southern passage shows that the detailed saddle bathymetry can have a strong influence on the flow of most dense Makarov Basin waters crossing the LR. Although, our survey does not fully resolve the hydrography and bathymetry over the southern saddle, it indicate that relatively minor topographical features may control if the dense waters reaching the saddle cross over the ridge or return back to the Makarov Basin.

**Technical corrections:**

We appreciate these detailed comments. We will go through the manuscript and address all suggested technical corrections.